# Effect of Supplementing a *Bacillus* Multi-Strain Probiotic to a Post-Weaning Diet on Nutrient Utilisation and Nitrogen Retention of Piglets

**DOI:** 10.3390/ani13233597

**Published:** 2023-11-21

**Authors:** Anne Maria Stevina Huting, Liz Vanessa Lagos, Lea Hübertz Birch Hansen, Francesc Molist

**Affiliations:** 1Schothorst Feed Research B.V., 8218 NA Lelystad, The Netherlands; ahuting@schothorst.nl (A.M.S.H.); vlagos@schothorst.nl (L.V.L.); 2Animal and Plant Health and Nutrition, Chr. Hansen Holding A/S, 2970 Hoersholm, Denmark; dkleha@chr-hansen.com

**Keywords:** *Bacillus amyloliquefaciens*, *Bacillus subtilis*, energy utilisation, ileum digestibility, volatile fatty acids

## Abstract

**Simple Summary:**

There is a global need to produce pork that is free of antimicrobials and has a minimum impact on the environment. Most antibiotics are used during the nursery phase mainly for gastro-intestinal (GIT) diseases. Nutrition can help support gut health during the challenging process of weaning. At weaning, piglets experience short-term anorexia and suffer from an impaired development of the gastrointestinal tract. In order to keep weaned piglets healthy, it is important to support the function and development of the GIT of piglets’ by enhancing nutrient digestibility and absorption. Especially since undigested proteins entering the large intestine are one of the major risk factors for the development of post-weaning diarrhoea. Probiotics (i.e., live microorganisms) are suggested to not only modulate piglets’ health but also enhance nutrient utilisation; therefore, they may contribute to improved piglet health. The aim of this study was to evaluate the effect of a novel *Bacillus* multi-strain on nutrient efficiency. This study demonstrated that the use of the multi-strain *Bacillus* probiotic (*B. amyloliquefaciens* and *B. subtilis*) was able to improve the nutrient efficiency of weaned piglets and may contribute to a reduced N pollution into the environment.

**Abstract:**

Probiotics are suggested to improve pig health, nutrient utilisation, performance, and they may reduce nitrogen (N) pollution. However, the effectivity of a single strain might be different from that of a multi-strain. The study was conducted to investigate the effect of a novel *Bacillus* multi-strain on nutrient digestibility, energy utilisation, and N retention in weaned piglets using an European diet. The experiment consisted of a control diet (CD) and a supplemented diet (SD). The probiotic used for SD consisted of *B. amyloliquefaciens*—516 and *B. subtilis*—541. A total of eight boars/treatment were weaned (day 0; 8.5 kg body weight). Only boars were used to ease the collection of urine. Until day 10, piglets were fed ad libitum and were housed in pairs; from day 11, piglets were fed semi ad libitum (feeding level 3.2× metabolic body weight) and were housed individually. From day 14, faecal and urine were collected twice daily. Piglets were humanely euthanised at day 19 (15.0 kg bodyweight) after which the jejunum, ileum, and colon content were collected. In faeces, the apparent total tract digestibility (ATTD) of, amongst others, DM, organic matter (OM), crude protein (CP), non-starch polysaccharides (NSP), and subsequently net energy (NE) were calculated using titanium dioxide as an indigestible marker. In the jejunum and ileum, the apparent digestibility of CP was estimated, and in the ileum, the apparent AA digestibility was measured. In urine, the N content was measured to determine N retention. The volatile fatty acid (VFA), branched-chain fatty acid (BCFA), and lactic acid content were analysed in the colon and faeces. The apparent CP digestibility in the jejunum and ileum was not affected by treatment (*p* > 0.05), and no effect was observed on the apparent ileal digestibility of AA (*p* > 0.05). Supplementation with the multi-strain probiotic improved the ATTD of DM (*p* = 0.01; +1.3%) and OM (*p* = 0.02; +1.2%) and tended to improve the ATTD of CP (*p* = 0.10; +2.2%) and NSP (*p* = 0.07; +1.9%). The multi-strain probiotic also improved the NE value (*p* = 0.02; +0.2 MJ/kg DM) and improved N retention (*p* = 0.05; +1.6%). Supplementation did not influence the VFA, BCFA, and lactic acid content in the faeces (*p* > 0.05). However, in the colon, supplementation did influence the lactic acid content (lower; *p* = 0.01) and tended to influence the valeric acid content (higher; *p* = 0.09). In conclusion, results from the current study suggest that the multi-strain probiotic has the potential to contribute to improve nutrient efficiency in weaned piglets. More research needs to be conducted to identify the impact of the improved nutrient utilisation on gut health in post-weaned pigs as well as environmental pollution.

## 1. Introduction

The various stressors at weaning have a great impact on piglet performance and health. This, along with restrictions with respect to the use of antimicrobials, have led to a significant shift in the development of bioactive alternatives. One of such alternatives may be probiotics (i.e., live microorganisms), which are suggested to not only modulate piglets’ health [1,2,3] but to also enhance nutrient utilisation [4,5] and ultimately performance [2]. Especially, the *Bacillus*-based probiotic has gained interest due to its superior characteristics to withstand harsh environmental conditions [2,4,6]. Furthermore, *Bacillus* strains have been suggested to produce various extracellular enzymes [6], which may influence nutrient utilisation. In particular, *Bacillus* strains *B. amyloliquefaciens*, *B. subtilis*, and *B. mojavensis* are suggested to have strong probiotic potentials [7]. Whether a probiotic is effective depends on the diet, strain, dose, and several pig factors including health status and age [8]. Additionally, the mechanism of action of a single-strain probiotic might be different from that of a multi-strain [5]. It is furthermore speculated that probiotics play a not only in digestion but also in absorption and propulsion in the gastrointestinal tract, which may influence nutrient retention [9]. Due to sustainability concerns, besides the effects on animal health and performance, there is interest in the role of probiotics in reducing nitrogen (N) pollution, but studies evaluating this are lacking. For instance, it is suggested that probiotics could possibly convert ammonia into bacterial proteins reducing excretion [10].

The objective of the current study was to evaluate whether a novel multi-strain probiotic consisting of *B. amyloliquefaciens* and *B. subtilis* could modify nutrient digestibility, energy utilisation, amino acid (AA) digestibility, and N retention in weaned piglets. Weaned piglets were used as probiotics are suggested to be more effective in animals with an impaired gastrointestinal tract (GIT) than in older pigs [11]. *Bacillus amyloliquefaciens* can synthesise protease, whereas *B. subtilis* is able to produce α-amylase [4,5]; therefore, it was hypothesised that the multi-strain probiotic improves nitrogen and energy efficiency in weaned piglets. This is important, as limiting excess nutrients passing the distal intestine is key to avoid osmotic diarrhoea as a result of harmful end-products produced by the fermentation of proteins (reviewed by Huting et al. [12]). The *Bacillus* strains of interest were already individually tested in a protein and amino acid digestibility trial in pigs [4]. However it is unknown whether mixing strains would result in an additive, synergistic, or even a reduced efficiency. Furthermore, in previous digestibility trials, the experimental diets used were suitable for the North American market and consisted mostly (i.e., >50%) of maize, soybean meal, and soy protein concentrate (SPC) [4,13,14,15], whereas in current study, the diets were formulated to meet the European market (<25% of the diet consists of maize, soybean meal and SPC). These differences in the dietary composition may influence the effectivity of the probiotic. It was therefore of interest to evaluate the effectivity of the strains combined on amongst others nutrient utilisation in weaned piglets.

## 2. Materials and Methods

The study was conducted at the research facility of Schothorst Feed Research B.V. (SFR, Lelystad, The Netherlands). The protocol of the experiment (AVD24600202010384) was approved by the Animal Care and Use committee of SFR (Lelystad, The Netherlands) and in accordance with the Dutch law on animal experimentation, which complies with the European Directive 2010/63/EU on the protection of animals used for scientific purposes.

### 2.1. Experimental Design and Diets

The experiment consisted of two dietary treatments: a control diet (CD) and a supplemented diet (SD). The CD was mainly based on wheat (35%), barley (20%), soybean meal (11%), maize (10%), and wheat middlings (10%) and was formulated to meet the Foundation Central Bureau for Livestock Feeding (CVB) recommendations for essential nutrients of weaned piglets (see Table 1). The SD diet was prepared as the CD basis but supplemented with the *Bacillus* multi-strain probiotic SOLPREME^®^ (Chr. Hansen A/S; Hoersholm, Denmark) at 1.1 × 109 CFU per kg diet (actual dose 0.04%). The multi-strain probiotic (minimal 2.75 × 109 CFU per g product) consisted of the viable spores *Bacillus amyloliquefaciens*—516 and *Bacillus subtilis*—541 and was supplied using a CaCO_3_ carrier. Therefore, less limestone (−4.7%) was added to the SD to obtain similar Ca levels to the CD. The experimental diets did not contain antibiotics, acidifiers or phytase, but they contained zinc and copper supplements at levels according with the European law for weaned piglets [16,17]. Titanium dioxide (0.5%) was added to the experimental diets as an indigestible marker to calculate nutrient digestibility.

The experimental diets were produced in the specialised feed mill of Research Diet Services (Wijk bij Duurstede, The Netherlands) and pelleted at a diameter of 3 mm. The temperature during diet production ranged between 75 and 76 °C for the pellet press. The CFU recovery was analysed in the mash and pelleted diets prior to the experiment (Chr. Hansen A/S; Hoersholm, Denmark). As a method of blinding, numerical coding was used. Each pen was identified with a diet code. Numerical-coded feed bags and pens matched treatment numerical codes.

### 2.2. Animals and Animal Housing

A total of 16 piglets (Tempo × TN70) were weaned at around 30 days of age (29.8 ± 1.11 days) with a weight of 8.48 ± 0.272 kg. Piglets were ear tagged at birth, and no teeth clipping, tail docking or castration was performed. The newly born piglets were injected with injectable iron (1 mL, Iron-ject^®^, Dopharma, The Netherlands) at 3–4 days of age. Piglets were not vaccinated pre- or post-weaning, but the progenitors (gilts and sows) were vaccinated according to the manufacturer’s vaccination scheme with an inactivated vaccine against neonatal colibacillosis and Clostridium infections (SUISENG^®^, HIPRA, Amer, Girona, Spain). Piglets received creep feed from approximately one week of age until weaning. The selected weaned piglets were free from signs of injury or illness. Only boars were used in this trial to ease the collection of urine, as it can be collected separately without being contaminated by faecal material as would be the case when using gilts. Each experimental treatment consisted of 8 replicates of boars, which were randomly allocated on the basis of weaning weight to the different treatments and pens. Piglets were blocked to replicates based on their weaning weight. Variation in weaning weight was 800 g with the lightest piglets weighing 8.10 kg (i.e., replicate 1) and the heaviest piglets weighing 8.90 kg (i.e., replicate 8).

The health status of the piglets was checked and recorded daily. Piglets were inspected at least once a day by an animal caretaker. If an animal was in poor condition, it was observed more frequently. If deemed unlikely to recover or survive, the animal was humanely euthanised. In case (antibiotic) treatment was necessary, the individual pig number, the kind of treatment, and treatment duration were recorded. In case of mortality, the cause of death was recorded. Faecal consistency (see for protocol Guan et al. [18]) was determined daily at the pen level between day 0 and 11 post-weaning and at the individual level between day 11 and 19 post-weaning.

#### 2.2.1. Animal Housing

Piglets were housed, fed, and managed according to directive 2010/63/EU for the protection of animals used for scientific purposes. Weaned piglets were housed in pairs in a total of 8 pens (2.00 × 1.00 m) from weaning (day 0) until day 11 post-weaning. The pen floor was partly slatted and partly covered with a rubber mat. Each pen was equipped with two feeding troughs (with 1–2 feeder places), two drinking nipples, a metal chain with MS pig play material (horizontal or vertical bars; MS Schippers, Hapert, The Netherlands), and a cotton rope (MS Schippers, Hapert, The Netherlands). At day 11 post-weaning, piglets were individually weighed and subsequently separated (i.e., individually housed) through the placement of a transparent wall in each pen. Piglets were housed individually for the remaining experimental period (until day 19 post-weaning; around 49 days of age). Final pen dimensions at individual housing were 1.0 × 1.0 m. Room temperature and relative humidity were recorded daily and were mechanically controlled by a climate computer following a temperature curve targeted at 29 °C at the day of weaning (27.9 ± 0.10 °C) to 25 °C at 19 days post-weaning (24.7 ± 0.33 °C). The rooms were ventilated using outdoor air. Humidity in the rooms was dependent of outdoor humidity and ventilation rate (humidity 63.0 ± 6.12%: range 51.3–76.6%; the trial was executed in June). Artificial lights were provided from 0630 until 1800 h.

#### 2.2.2. Feeding Scheme

From weaning until day 10 post-weaning, piglets were fed the experimental diet ad libitum. Piglets were individually weighed at day 11 post-weaning for the estimation of the feeding portion. From day 11 post-weaning onwards, piglets were fed following a semi-ad libitum feeding scheme (3.2× metabolic body weight), which was calculated by the following formula (adapted from [19]):(1)$$\mathrm{Semi}\;\mathrm{ad}\;\mathrm{libitum}\;\mathrm{feeding}\;\mathrm{scheme}=\frac{0.419\times {BW}^{0.75}\times 0.7\times 3.2}{NE}$$
where 0.419 is the energy for maintenance per kg metabolic body weight (BW); BW^0.75^ is the metabolic body weight; 0.7 is the factor to calculate net energy (NE) from metabolic energy (ME); and 3.2 is the feeding level used: in this case, 3.2× maintenance level for NE. The feeding portion increased daily by approximately 3% on an estimated growth curve. The average daily feed intake during individual housing was 672 ± 72.7 g/day.

The feed was spread over 2 feeding portions (i.e., morning and afternoon) from day 11 until day 16 post-weaning. On day 17 and 18 post-weaning, the total feeding portion was split into 6 smaller portions and was fed every 2.5 h (between 0600 and 1830 h) in order to reach a steady state. On the day of euthanasia, piglets were fed half the daily feed portion spread over 3 smaller portions: 1/6 portion 6 h before euthanasia, 1/6 portion 3.5 h before euthanasia, and 1/6 portion 1 h before euthanasia in order to ensure all parts of the digestive system were filled with sufficient content. Drinking water was available ad libitum throughout the trial.

### 2.3. Sampling and Analytical Methods

During the last 5 days of the experimental period (from day 14 post-weaning onwards), faecal samples were collected twice daily (830 and 1500 h) from the pens. Urine was collected via a funnel from the tray underneath the pen into a bucket (total collection). Hydrochloric acid was added to urine after each sampling time (5 mL/time; adjusted to the actual volume of urine that was collected). Faecal and urine samples were stored at 4 °C during the collection period. Faeces and urine were stored at −20 °C and freeze-dried for further analysis. Piglets were weighed and humanely euthanised via intracardiac injection with T61^®^ (MSD Animal Health, Boxmeer, The Netherlands) after sedation with Zoletil^®^ (Virbac, Barneveld, The Netherlands) to facilitate the collection of the digesta at day 19 post-weaning (15.2 ± 1.43 kg). During dissection, the entire small intestine was spread on a table. The last 2 m from 2/3 of the small intestine was considered to represent the jejunum, and the distal 2 m from the small intestine was considered to represent the terminal ileum. The mentioned segments were dissected to collect its content by gentle stripping. The digesta content was homogenised by manual mixing, and pH was immediately measured with a portable pH meter (Mettler-Toledo B.V., Tiel, The Netherlands). Digesta samples were frozen at −20 °C and freeze-dried for further analysis. The dried faecal and digesta samples were milled through a 1 mm sieve prior to chemical analyses.

Fresh colon (i.e., 1 g collected 1 m from the cecum) and faecal samples (i.e., 1 g fresh material) were collected and stored at −20 °C for further analysis. Volatile fatty acids (VFAs: the sum of acetic acid, propionic acid, and butyric acid), valeric acid, branched-chain fatty acids (BCFAs: the sum of Iso-butyric, 2-Methyl-butyric, and Iso-valeric), and lactic acid content in the colon and faecal samples were analysed in mmol/kg by BaseClear B.V. (Leiden, The Netherlands). Lactic acid, VFAs, and BCFAs were derivatised to the respective phenyl esters by using phenyl chloroformate reagent. The resulting esters were analysed by Agilent GC-FID. Matrix-matched internal standard calibration with butyric-d7—and acetic-d3 acids was used in quantitation.

The chemical analyses were performed in duplicate. Dry matter (DM) content was determined by drying to constant weight at 103 °C [20]. Crude protein (CP; nitrogen × 6.25; [21]) was determined by combustion according to the Dumas principle; the crude fat (CFat) content was determined using ether extraction after hydrolysis with hydrochloric acid under heating; crude ash was measured gravimetrically after ashing the sample for 3 h at 550 °C [22]; starch content was determined enzymatically according to NEN-ISO 15914:2005EN [23]; sugar was measured according to [24] (ANAL-10138; NutriControl B.V.); amino acid (AA) content was measured according to [24] (ANAL-10018; NutriControl B.V.), tryptophan content was measured according to [24] (ANAL-10017; NutriControl B.V.), and titanium content was measured as described by Short et al. [25]. The titanium (Ti) content as found in the experimental diets was as expected (ranging 2.95–3.01 g/kg; expected content was 3.00 g/kg), the titanium content as found in the jejunum was 4.55 g/kg (SD = 1.059), which was 7.92 g/kg in the ileum (SD = 1.819) and 17.2 in the faecal contents (SD = 1.03).

### 2.4. Calculations and Statistical Analysis

The organic matter (OM) content and non-starch polysaccharides (NSP) were calculated in accordance with CVB [26]. The digestibility coefficients (DC, %) of nutrients were calculated using the following equations:OM = DM − Ash(2)
NSP = OM − CP − CFat − Starch − CF_di_ × Sugar(3)
Marker_ratio_ = Ti_diet_/Ti_excreta_(4)
DC_nutrient_ = (1 − Marker_ratio_ × Excreta_nutrient_/Diet_nutrient_) × 100%(5)
where CF_di_ is the correction factor to convert the analysed sugar content (expressed as glucose equivalent) into sugar, and the marker is the indigestible marker content (measured as Ti in g/kg DM); excreta is defined as faecal, proximal jejunal digesta, mid-jejunal digesta, or ileum digesta; DC is the apparent digestibility coefficient (in %); and nutrient is the content of AA, CP, CFat, OM, DM, or ash (in g/kg DM) in the diet and the excreta. For the calculations of the jejunal digesta and ileum digesta, everything was expressed in g/kg. The NE value was calculated in accordance with CVB [26] following:NE value (in MJ/kg DM) = 11.7 × DC_CP_ × CP + 35.74 × DC_CFat_ × CFat + 14.14 × Starch + 12.726 × Sugar × DC_sugar_ + 9.74 × DC_NSP_ × NSP(6)
where DC is the is the apparent total tract digestibility (ATTD) coefficients as found in the current study (in %) and the nutrient content (i.e., CP, CFat, starch, sugar, and NSP) is given in g/kg DM. For starch, a 100% ATTD was assumed, and for sugar, the enzymatically digestible sugar (DC_sugar_ in %) content of the experimental diets was estimated based on CVB [26]. The nitrogen (N) retention and biological value (following [4]) were calculated using the following equations:N_retention_ (in g) = (N_intake_ × ATTD_N_) − (N_urine_ × Total_urine_)(7)
N_retention coefficient_ (in %) = (N_retention_/N_intake_) × 100(8)
Biological value (in %) = (N_retention coefficient_/(N_intake_ − N_faeces_)) × 100(9)
where N_intake_ is the total N intake (in g); ATTD_N_ is the apparent total tract digestibility of N; N_urine_ is the N content in urine (in g/kg); Total_urine_ is the total urine production (in kg); and N_faeces_ is the amount of N that was found back in the faeces (undigested).

Based on the sample size calculation with GenStat^®^ for Windows Version 21 (VSN International Ltd., Hemel Hempstead, UK) using the ASAMPLESIZE procedure with a significant level of α < 0.05 and a power of 0.80, a total of 8 piglets per treatment were needed. For estimating the sample size, the ATTD CP digestibility was considered as one of the important outcome parameters. Based on the study of Jørgensen et al. [11], a difference of 2.5% in ATTD CP was found with an SEM of 0.38 (12 pigs/treatment). Based on the study of Lewton et al. [5], a difference of 18.4% was found in distal colon N digestibility with an SEM of 4.22 (6 piglets/treatment). The Shapiro–Wilk test was used to test for the normal distribution of residuals (WSTATISTIC procedure), and Bartlett’s test was used to test for homogeneity of variances (VHOMOGENEITY procedure). The ABOXCOX procedure was used in case data were not normally distributed and needed to be transformed. For presentation purposes, the calculated means were back-transformed and are presented together with the 95% confidence interval (CI, using Bonferroni inequality) instead of SEM. The experimental results were analysed using the TTEST procedure by GenStat^®^. For all parameters except performance between day 0 and 11 post-weaning, the experimental unit was piglet (*n* = 8); for performance data between day 0 and 11 post-weaning, pen was the experimental unit (*n* = 4). Replicate was used as a random block effect. For the digestibility coefficients data, an extra quality check was performed: if the dietary OM digestibility of the animal differed by more than 2.5 times the standard deviation from the average, the piglet was considered an outlier (based on CVB protocol for digestibility studies [19]). If this was the case, all the digestibility data (e.g., also at the intestinal level) of this animal were excluded from the analysis. All data were screened for outliers; data were identified as outliers if the residual (fitted-observed value) differs > 2.5 × standard error on the residuals of the data set. In addition, if this was the case for CP digestibility or CFat digestibility, the digestibility coefficient for the NSP and NE values was also regarded as an outlier [19]. Missing values were estimated through GenStat^®^ (using least square estimates). Treatment means were compared using the Least Significant Differences (LSD, Fisher’s LSD method). A T-probability of *p* ≤ 0.05 was considered statistically significant, while 0.05 < *p* ≤ 0.10 indicated a near-significant trend. The data are presented as means ± SEM.

## 3. Results

The analysed nutrient contents of the experimental diets are presented in Appendix A. The CFU recovery was analysed in the mash and pelleted diets and was in line with expected values (Appendix A).

### 3.1. Exclusion of Animals

No pen or piglet was considered an outlier on the basis of the performance data. All piglets passed the quality check following CVB [19] protocol, which means that the individual dietary ATTD of OM did not differ by more than 2.5 times the standard deviation from the average of the particular treatment. Piglets from replicate 2 (i.e., one piglet from the CD and one piglet from the SD treatment) had an ileum pH content that was >2.5 times lower than the standard error of the residuals and were therefore considered outliers. Replicate 2 from treatment SD did not have enough ileal content to perform the AA analysis; therefore, this piglet was treated as a missing value in the subsequent analysis.

### 3.2. Animal Performance and Health

Piglets weighed on average 11.3 kg (SD = 1.04) at day 11 post-weaning and 15.2 kg (SD = 1.43) at day 19 post-weaning. Experimental treatment did not influence piglet performance or faecal score (Appendix A). No piglets died during the course of the trial. One piglet from the CD treatment was treated with painkillers (i.e., Ketoprosol 10%, AST Farma, Raamsdonksveer, The Netherlands) for lameness at day 12 and day 17 post-weaning.

### 3.3. Apparent Small Intestine Digestibility Coefficients

The effect of dietary treatment on apparent nutrient digestibility in the small intestine can be found in Table 2. The apparent CP digestibility in both the jejunum and ileum was not affected by dietary treatment (*p* = 0.18 and *p* = 0.32, respectively). The effect of dietary treatment on apparent ileal digestibility (AID) of AA and digesta pH can be found in Appendix A, respectively. The results indicate that dietary treatment did not significantly influence the AID of AA nor the pH of the digesta content.

### 3.4. Apparent Total Tract Digestibility Coefficients and NE Value

The effect of dietary treatment on ATTD coefficients and NE value can be found in Table 2. Supplementation with the *Bacillus* multi-strain probiotic did not improve the ATTD of CFat (*p* = 0.60) and ash (*p* = 0.17). Supplementation with the *Bacillus* multi-strain probiotic influenced the ATTD coefficients of DM (*p* = 0.01) and OM (*p* = 0.02) and tended to influence the ATTD coefficients of CP (*p* = 0.10; Δ2.2%) and NSP (*p* = 0.07; Δ1.9%). Weaned piglets fed the supplemented diet had a 1.3% higher ATTD coefficient of DM and a 1.2% higher ATTD coefficient of OM. Next to the improvements in nutrient digestibility, supplementation with the *Bacillus* multi-strain also increased the NE value with 0.2 MJ/kg DM (*p* = 0.02) compared with the CD treatment.

### 3.5. N Retention

The effect of dietary treatment on N retention can be found in Table 3. Supplementation with the multi-strain probiotic did not influence N intake (*p* = 0.81), faecal N content (*p* = 0.11), digestible N content (*p* = 0.95), urinary N (*p* = 0.79), N retention when expressed in g (*p* = 0.99), or the biological value (*p* = 0.76). However, supplementation with the multi-strain probiotic improved N retention (*p* = 0.05) with the SD treatment resulting in a 1.6% higher N retention coefficient than the CD treatment.

### 3.6. VFAs, BCFAs, and Lactic Acid Content in the Colon Digesta and Faecal Material

The effect of dietary treatment on VFAs, BCFAs, and lactic acid content in the colon and faeces can be found in Table 4. No effects (*p* > 0.05) were found at the faecal level with respect to VFAs, BCFAs, and lactic acid content. However, in the colon, dietary treatment did influence the lactic acid content *(p* = 0.01) and tended to influence the valeric acid content (*p* = 0.09). Supplementation with the *Bacillus* multi-strain probiotic resulted in a higher colonic valeric acid content (+1.74 mmol/kg) and a lower lactic acid content (−0.21 mmol/kg).

## 4. Discussion

This paper describes a study in which the effect of the multi-strain probiotic consisting of *B. amyloliquefaciens* and *B. subtilis* on nutrient digestibility, energy utilisation, and AA digestibility in weaned piglets is evaluated. It was hypothesised that the *Bacillus* multi-strain could improve nitrogen utilisation by *B. amyloliquefaciens* that synthesise protease and energy efficiency by *B. subtilis* that synthesise, amongst others, α-amylase and fibre-degrading enzymes [4,5]. An improved N utilisation is important for piglets’ health [12] but also for reducing N pollution. Increasing N utilisation reduces urinary and total N excretion, further reducing N pollution. This can be manipulated through the diet by for instance reducing the CP content, using highly digestible feedstuffs, synthetic amino acids, or zootechnical additives that can improve the utilisation of CP [27]. To understand the potential of the *Bacillus* multi-strain in reducing N excretion, N balance was evaluated in this trial.

The current study was not set up to evaluate the effect of the *Bacillus* multi-strain on post-weaning performance. The four pens (with only two piglets/pen) per treatment during the ad libitum feeding period were not expected to provide enough power to find differences in performance. However, the eight piglets/treatment provided enough statistical power to detect differences in nutrient utilisation.

Part of the large between-animal variation as observed in the current study and perhaps the lack of significant effects at the small intestinal level might be explained by the method used. In the current study, the samples were obtained using the slaughter technique, in which samples can only be obtained once compared with, for instance, the T-cannula where samples are obtained for a prolonged period of time [28]. Nonetheless, to limit the shedding of the intestinal cells and mucus into the digesta, the piglets were euthanised under sedation, and squeezing of the intestinal tract was avoided [29]. In addition, it should be noted that one piglet from the CD treatment was treated twice with Ketorosol for lameness (i.e., day 12 and 17 post-weaning). Ketoprosol is a Non-Steroidal Anti-Inflammatory Drug (NSAID), which is suggested to alter the composition of the microbial community of elderly that frequently use NSAIDs (i.e., ≥3 times a week) [30]. They furthermore speculated that an altered microbiota could influence nutrient utilisation [30]. While in the current trial, we did not have profound reasons to exclude this piglet on the basis of the medical treatment it required (e.g., the animal was not considered an outlier), the conclusions of the trial would not change after running the model excluding the piglet. Therefore, in the present experiment, it was decided to not exclude this piglet.

Using canulated growing-finishing pigs, the supplementation of *B. amyloliquefaciens*, but not *B. subtilis*, improved the AID digestibility of some dispensable and indispensable AA, though in the current study, no such effects were found [4]. Also, Lewton et al. [5] found no improvement after the administration of probiotics on the AA digestibility in the ileum and only found significant effects in the jejunum (approximately 8 m proximal from the cecum). It was suggested that as *Bacillus* is a member of the firmicutes phyla, it may have helped to restore the microbiota composition of the jejunum generally being *Firmicutes*-dominated [5] and therefore may have contributed to an increased nutrient utilisation in the jejunum. Although the results of the current study did not show significant effects, it suggests that the *Bacillus* multi-strain was able to numerically improve apparent CP digestibility in the proximal intestine (i.e., +7%).

The observation that supplementation with the *Bacillus* multi-strain probiotic tended to increase the ATTD of CP (+2.2%) and resulted in a numerically lower faecal N content (−2.8 g) is in agreement with other studies using nursery pigs [13,14,15]. It is speculated that the improvement in N utilisation might be a result of (1) the metabolites produced by the probiotic that enhances nutrient digestibility; (2) improved gut development supporting digestion and absorption of nutrients; and (3) its effect on gut microbiome composition which alters gut health [15]. The absence of a significant effect may be amongst others a result of differences in the diet composition. Probiotics may have a greater potential when the inherent digestibility of the diet is already relatively high [31], when the CP content of the diets are low [5], or when supplementing low-energy diets [11]. In the studies of Giang et al. [13], Lee et al. [14], and Cai et al. [15], the CP content of the experimental diets was >20%, whereas in the current study, this was considerably lower (<18%); the NE content was not different between studies (i.e., around 10–11 MJ/kg DM). The diet in the current study mostly consisted of cereal (by-products) and soybean meal (75% and 11%, respectively), whereas in the other studies [13,15], the inclusion of highly digestible protein sources was higher, such as milk products (e.g., whey powder, sweet whey, milk replacer), soy protein concentrate, and processed feedstuffs (i.e., extruding or fermentation). Nonetheless, the AID and ATTD values of CP of the control diet in the current study were, respectively, 69% and 81%; this was slightly lower than Giang et al. [13] (71–75% and 83–84%, respectively) but higher than the ATTD of CP that was found by Cai et al. [15] (78%) and Lee et al. [14] (72%). Thus, differences in diet composition cannot entirely explain the results, but perhaps also factors like the strain and the dose used may play a role. Additionally, it has to be noted that piglets in the current study were on average 30 days of age at weaning with a weaning weight of 8.5 kg, whereas in the study of Lee et al. [14], the piglets were 21 days old and 6.4 kg; this was 24 days and 6.8 kg, respectively, for Cai et al. [15], and 27 days and 7.7 kg, respectively, for Giang et al. [13]. It is therefore speculated that the piglets used in the current study might be more robust (i.e., older and heavier at weaning) and had a less challenging diet with respect to CP level [12] than the piglets used in the other studies. This may have contributed to the lack of significant effect. For instance, it has been suggested that probiotics are more effective in animals with an impaired GIT (e.g., unstable microbiota) and not so much for older pigs when pigs are morecapable of resisting intestinal disorders like for instance during the finisher phase [11]. However, more research is necessary to evaluate whether the multi-strain probiotic would be more beneficial in improving nutrient utilisation when piglets are reared under more challenging conditions.

Organic matter is the calculated fraction of CP, CFat, and carbohydrates (i.e., the sum of sugars, starch, and NSP). The significant improvement of ATTD coefficients of OM as found in the current study as a result of the multi-strain probiotic might therefore be mostly a result of the numerical higher CP digestibility (+2.2%) and NSP digestibility (+1.9%). The fact that a higher ATTD of DM (+1.3%) was found for weaned piglets fed the diet containing the multi-strain probiotic might be a result of the significant improvement of the ATTD of OM (+1.2%) and the numerically higher ATTD of ash (+1.4%). However, the differences in ATTD DM seems rather small compared with other studies that found an ATTD DM improvement ranging between 1.9 and 3.5% [14,27,31]. Furthermore, in the current study, the multi-strain probiotic improved NE utilisation. Other studies [4] found an increased energy utilisation in pigs after supplementation with *Bacillus* strains. The difference in energy utilisation in the current study may come from the numerical improvements in the ATTD of CP and NSP. On the other hand, it is suggested that some strains of *Bacillus* can synthesise α-amylase and fibre-degrading enzymes [4]. The enhanced fermentation of dietary fibre can increase the production of volatile fatty acids (VFAs) and subsequently increase energy utilisation [32]. It is suggested that the colonic fermentation of NSP results in VFA and lactic acid contributes to approximately 20% of the total dietary energy utilisation in adult pigs [33]. The calculated soluble (usually fermentable) carbohydrate content of the experimental diets (i.e., hemicellucose and pectins) was 102 g/kg, whereas the insoluble (usually inert) carbohydrate content (i.e., lignin and cellulose) was 63 g/kg. Piglets, however, only have a limited fermentation capacity, and therefore, the inclusion of soluble carbohydrates should be limited in weaned piglet diets [12]. The VFA molar proportion (i.e., acetate: propionate: butyrate) of the colon content and faecal content in the current study (i.e., 63:26:10 and 61:27:12, respectively) was in line with Jaworski et al. [32]. However, similarly to Jaworski et al. [32], the addition of the multi-strain probiotic did not influence the total VFA content. The absence of effect might be a result of the absorption of VFA in the cecum, which is rather efficient, and therefore measuring the VFA content in the colon and faeces might not be accurate [34], and using in vitro methods might be more applicable [33]. On the other hand, in the present study, weaned piglets fed a diet with *Bacillus* strains tended to have a higher colonic valeric acid content and a significant lower colonic lactic acid content. The slightly higher colonic valeric acid content in the supplemented treatment may suggest a higher hindgut fermentation of protein [29]. However, (1) the total BCFA content formed by the fermentation of branched-chain amino acids (BCAAs) was not significantly affected; and (2) the AID values of CP and BCAAs (i.e., valine, leucine, and isoleucine) were not affected by dietary treatment, suggesting that no more undigested protein entered the large intestine to be fermented. These results can therefore not explain the higher colonic valeric acid content after supplementation, and it is questionable whether the observed differences are large enough to have a biological relevance. The effects of *Bacillus* strains on organic acid concentrations (including lactic acid) in the intestinal lumen are inconsistent [13], but too high lactic acid contents may in fact be harmful for the pig [1]. The absence of differences in VFA production might also explain why in the current study no differences in digesta pH at the different segments along the GIT were found. It is also worth mentioning that it cannot be ruled out that supplemented pigs may have had a reduced maintenance energy requirement [32], which may have contributed to the improved energy efficiency. Thus, on the basis of the results of the present study, it is suggested that the multi-strain probiotic did not influence the production of SCFA, which is in contrast with others [10].

The observation that N retention expressed in g was not significantly influenced by the experimental treatment was probably a result of the relatively large SEM and variation in feed intake among piglets, for which the N retention coefficient was corrected. These results are in contrast to those of Blavi et al. [4], where no effects of single-strain supplementation of *B. amyloliquefaciens* and *B. subtilis* were observed on N retention. This might suggest a synergistic effect of the two strains in N retention or that the probiotic is more effective in younger piglets. However, more research is necessary to evaluate the effect of the multi-strain on N retention in weaned piglets.

## 5. Conclusions

In conclusion, the results from the current study suggest that the multi-strain probiotic consisting of the viable spores *B. amyloliquefaciens* and *B. subtilis* have the potential to improve nutrient efficiency in weaned piglets fed a European diet. Furthermore, it seems that the multi-strain probiotic could potentially contribute to a reduced N pollution. More research needs to be conducted to identify the impact of the improved nutrient utilisation on gut health in post-weaned pigs as well as environmental pollution.

## Figures and Tables

**Table 1 animals-13-03597-t001:** Diet composition and calculated nutrient values of the experimental diets including the control diet (CD) and the supplemented diet (SD).

	Unit	CD	SD
Wheat	%	35.0	35.0
Barley	%	20.0	20.0
Soybean meal	%	11.3	11.3
Maize	%	10.0	10.0
Wheat middlings	%	10.0	10.0
Soybean oil	%	2.70	2.70
Potato protein	%	2.50	2.50
Whey protein delactose	%	2.00	2.00
Soy protein concentrate	%	1.00	1.00
Molasses	%	1.00	1.00
Monocalcium phosphate	%	1.06	1.06
Salt	%	0.64	0.64
Lysine HCL (79%)	%	0.45	0.45
Methionine L/DL (99%)	%	0.15	0.15
Threonine L (98%)	%	0.14	0.14
Tryptophane L (98%)	%	0.03	0.03
Calcium formate	%	0.10	0.10
Titanium dioxide	%	0.50	0.50
Limestone	%	0.89	0.85
Vitamin/mineral premix ^1^	%	0.50	0.50
Copper sulphate (99%)	%	0.05	0.05
Probiotic	%	0.00	0.04
Moisture	g/kg	117	117
Ash	g/kg	59.3	59.3
Crude protein	g/kg	178	177
Crude fat (acid hydrolysis)	g/kg	52.9	52.9
Crude fibre	g/kg	31.8	31.8
Starch (enzymatic)	g/kg	393	393
Sugar	g/kg	43.2	43.2
NSPs	g/kg	156	156
Digestible Ca	g/kg	6.50	6.50
Net Energy (NE)	MJ/kg	9.81	9.81
SID Lys	g/kg	11.0	11.0
SID LYS/NE	ratio	1.15	1.15
SID Met/SID Lys	ratio	0.36	0.36
SID M + C/SID Lys	ratio	0.59	0.59
SID Thr/SID Lys	ratio	0.63	0.63
SID Trp/SID Lys	ratio	0.20	0.20
SID Val/SID Lys	ratio	0.67	0.67
SID Ile/SID Lys	ratio	0.57	0.57
SID Leu/SID Lys	ratio	1.06	1.06
SID Arg/SID Lys	ratio	0.83	0.83
SID His/SID Lys	ratio	0.35	0.35
SID Phe/SID Lys	ratio	0.69	0.69

^1^ Vitamin and mineral premix was added to provide the following nutrients per kg of diet: Vitamin A: 10,000 IU; Vitamin D_3_: 2000 IU; Vitamin B_1_: 1.0 mg; Vitamin B_2_: 4.0 mg; Niacin: 30 mg; D-pantothenic acid: 15 mg, Vitamin B_6_: 1.5 mg; Choline chloride: 150 mg; Biotin: 0.05 mg; Folic acid: 0.4 mg; Vitamin B_12_: 20 µg; Vitamin E: 40 mg; Vitamin K_3_: 1.5 mg; Cu: 20 mg; Fe: 100 mg; Mn: 30 mg; Zn: 70 mg; I: 0.7 mg; Se: 0.25 mg.

**Table 2 animals-13-03597-t002:** The effect of experimental treatment (i.e., control diet = CD; supplemented diet = SD) on apparent nutrient digestibility.

Parameter	CD	SD	SEM	*p*-Value
Small intestine, %				
Jejunal CP	37.0	44.0	3.34	0.18
Ileal CP	69.1	59.7	6.22	0.32
Total tract, %				
DM	80.7 ^a^	82.0 ^b^	0.26	0.01
Ash	53.0	54.4	0.66	0.17
OM	83.3 ^a^	84.5 ^b^	0.27	0.02
CP	80.9 ^x^	83.1 ^y^	0.79	0.10
CFat	72.5	73.2	0.96	0.60
NSP	52.1	54.0	0.63	0.07
NE value, MJ/kg DM	10.8	11.0	0.03	0.02

^a,b^ Values within a row with different superscripts differ significantly at *p* < 0.05. ^x,y^ Values within a row with different superscripts tended to differ at *p* < 0.10.

**Table 3 animals-13-03597-t003:** The effect of experimental treatment (i.e., control diet = CD; supplemented diet = SD) on nitrogen (N) retention.

Parameter	CD	SD	SEM	*p*-Value
N intake, g	112	109	6.5	0.81
Faecal N, g	21.2	18.3	1.09	0.11
Digestible N, g	90.5	91.9	5.81	0.95
Urinary N, g	7.69	8.15	1.171	0.79
N retention, g	82.8	82.9	4.91	0.99
N retention coefficient, %	74.2 ^a^	75.8 ^b^	0.47	0.05
Biological value, %	83.3	86.3	6.34	0.75

^a,b^ Values within a row with different superscripts differ significantly at *p* < 0.05.

**Table 4 animals-13-03597-t004:** The effect of experimental treatment (i.e., control diet = CD; supplemented diet = SD) on the colonic and faecal SCFAs, VFAs, and BCFAs content ^1^.

Parameter ^2^	CD	SD	SEM	*p*-Value
Colon, mmol/kg				
VFA content				
Acetic acid	93.0	92.6	0.90	0.77
Propionic acid	40.6	36.9	1.69	0.17
Butyric acid	16.6	14.3	1.05	0.16
Total VFAs	150	144	2.5	0.11
BCFA content				
Valeric acid	3.18 ^x^	4.92 ^y^	0.628	0.09
Total BCFAs	7.17	8.55	0.599	0.15
Lactic acid	8.17 ^b^	7.96 ^a^	0.039	0.01
Total	166	160	2.4	0.17
Faecal, mmol/kg				
VFA content				
Acetic acid	74.4	74.9	2.95	0.90
Propionic acid	34.3	31.2	1.55	0.20
Butyric acid ^3^	16.2 (−25.9–34.5)	14.8 (−26.7–33.9)	-	0.46
Total VFAs	124	121	5.0	0.62
BCFA content				
Valeric acid	3.61	3.55	0.151	0.79
Total BCFAs	11.9	11.8	0.52	0.86
Lactic acid	8.89	8.76	0.053	0.15
Total	145	141	5.1	0.61

^a,b^ Values within a row with different superscripts differ significantly at *p* < 0.05. ^x,y^ Values within a row with different superscripts tended to differ at *p* < 0.10. ^1^ VFAs = volatile fatty acids, which consists of the sum of acetic acid, propionic acid, and butyric acid; BCFAs = branched-chain fatty acids consisting of the sum of Iso-butyric, 2-Methyl-butyric, Iso-valeric, and valeric acid; SCFAs = short-chain fatty acids. ^2^ The experimental unit was piglet (*n* = 8). Replicates (1 to 8) were used as the random effect. The experimental results were analysed using a two-way analysis of variance (ANOVA) by GenStat^®^. ^3^ This parameter was considered not normally distributed in its original form (i.e., Shapiro–Wilk *p* < 0.05). Transformation suggestions were made by the “ABOXCOX” procedure in Genstat. The butyric acid values in the faeces were transformed using X^2^. For presentation purposes, the calculated means were back-transformed and are presented together with the 95% confidence interval (CI, using Bonferroni inequality) instead of SEM.

## Data Availability

The data presented in this study are available on request from the corresponding author. The data are not publicly available due to its proprietary nature.

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
