# Peer review of "Effect of Supplementing a Bacillus Multi-Strain Probiotic to a Post-Weaning Diet on Nutrient Utilisation and Nitrogen Retention of Piglets"

_animals, 2023, doi:10.3390/ani13233597_

Round 1
Reviewer 1 Report
Comments and Suggestions for Authors
The manuscript submitted for review is very interesting and provides a new perspective on the use of probiotics in pig nutrition. As a reviewer, I have a few comments and questions for the authors of the manuscript.
Simple summary: In my opinion, it requires a broader presentation of the problem for people who do not know the topic. I would like to ask you to expand this abstract.
Abstract: The information presented in the abstract is concise and presents the reason for undertaking the research, briefly the material and methods, as well as the results and conclusion.
I would like to hear from the authors why they chose boars and not gilts or pigs?
Introduction: The introduction of the work presents a short description of probiotics and stress factors on the health and production results of young pigs. I lack a clearly defined research hypothesis and research goal. There is also no explanation of how nitrogen can be reduced after using probiotics.
Material and methods: The authors present in great detail individual elements regarding the material and methods of the experiment. The material and methods are correctly presented and described. This is the strong point of the job.
Has the use of granulation had a negative impact on the probiotic added to the supplementation diet?
Why was it decided to conduct the experiment on young piglets and not on older pigs?
Results: The presented results clearly and clearly present the results of the experiment.
Discussion: The results were properly discussed and justified in the discussion.
Conclusion: Conclusions briefly and correctly describe the results obtained at work.
Author Response
We kindly thank you for reviewing our manuscript, your fruitful suggestions, and kind words. We clarify the revisions per comment below.
We have added the following to the simple summary in L12-18: “Most antibiotics are used during the nursery phase, mainly for gastro-intestinal diseases. Nutrition can help to support gut health during the challenging process of weaning. At weaning piglets experience short-term anorexia and suffer from an impaired development of the gastrointestinal tract. In order to keep weaned piglets healthy, it is important to support piglets nutrient digestibility and absorption. Especially since undigested proteins entering the large intestine are being one of the major risk factor for the development of post-weaning diarrhoea.” and the following to L19-20: “…may contribute to improved piglet health. The aim of this study was to evaluate the effect of a novel Bacillus multi-strain on nutrient efficiency.”
Boars were used in this study as from boars urine can be more easily collected separately without it being contaminated by faecal material as would be the case when using gilts. This has now been further explained in L132-133 by adding the following: “…as it can be collected separately without being contaminated by faecal material as would be the case when using gilts”. In the abstract the following was added to L30 “Only boars were used to ease the collection of urine.”
We have added the following to L70-72: “For instance, it is suggested that probiotics could possibly convert ammonia into bacterial proteins reducing excretion [10].”. The hypothesis of this research was further described in paragraph L73-92.
We believe that pelleting nor granulation did not influence the activity of the probiotics as indicated in Supplementary Table S2 illustrating the CFU recovery rate of the mash and pelleted diets.
The following was added to L76-77 in the introduction “Weaned piglets were used as probiotics are suggested to be more effective in animal with an impaired gastrointestinal tract (GIT) than in older pigs [11].”
Thank you for the kind words about the materials and methods, the results, the discussion and the conclusion.
Reviewer 2 Report
Comments and Suggestions for Authors
The manuscript by Anne M.S. Huting et al. describes work about supplementing a Bacillus multi-strain probiotic to a post-weaning diet on nutrient utilisation and nitrogen retention of piglets. There are quite a few places to be improved for further consideration.
1, The novelty of the current study should be further highlighted.
2, The method for NE calculation needs literature support (Lines 158-159).
3, The effect size should be given to confirm the sample size of 8 being correct (Lines 230-233).
4, There were only two groups in the current study, t-test should be used instead of ANOVA. So the statistical analysis was wrong. I cannot go further for the results.
Some minor issues the authors may want to consider:
Lines 271-273: It mentioned that one piglet from the CD treatment was treated with painkillers, but this was not discussed in the discussion section.
Line 313: In Table 4, most of the results were not statistically significant, but you spent some length discussing them in the discussion section and did not draw meaningful conclusions.
Author Response
We kindly thank you for reviewing our manuscript and your fruitful suggestions. We clarify the revisions per comment below.
We have added the following to L85-90: “Furthermore, in previous digestibility trials the experimental diets used were suitable for the North American marked and consisted mostly (i.e. >50%) of maize, soybean meal, and soy protein concentrate (SPC) [4,13–15], whereas in current study the diets were formulated to meet the European market (<25% of the diet consists of maize, soybean meal and SPC). These differences in the dietary composition may influence the effectivity of the probiotic.”
The reference for the semi ad libitum feeding scheme has been added in L177. The NE calculation is clarified in L244-247.
We have added the following to L257-261: “For estimating the sample size the ATTD CP digestibility was considered as one of the important outcome parameters. Based on the study of Jørgensen et al [11] a difference of 2.5% in ATTD CP was found with a SEM of 0.38 (12 pigs/treatment). Based on the study of Lewton et al [5] a difference of 18.4% was found in distal colon N digestibility with a SEM of 4.22 (6 piglets/ treatment).”
A very good point regarding the statistical analysis. Although in the manuscript it was stated that ANOVA was used, but the actual test that was used was the TTEST procedure. This has now been clarified in L267.
We have added the following to L381-390: “In addition, it should be noted that one piglet from the CD treatment was treated twice with Ketorosol for lameness (i.e. day 12 and 17 post-weaning). Ketoprosol is a Non Steroïdal Anti-Inflammatory Drug (NSAID) which is suggested to alter the composition of the microbial community of elderly that frequently use NSAIDs (i.e. ≥3 times a week) [25]. They furthermore speculated that an altered microbiota could influence nutrient utilization [25]. While in current trial we did not had profound reasons to exclude this piglet on the basis of the medical treatment it required (e.g. the animal was not considered and outlier), the conclusions of the trial would not change after running the model excluding the piglet. Therefore in the present experiment it was decided to not exclude this piglet.”
We have added the following to L480-482: “Thus, on the basis of the results of the present study it is suggested that the multi-strain probiotic could not influence the production of SCFA which is in contrast with others [10].”
Reviewer 3 Report
Comments and Suggestions for Authors
The study was well designed but the paper as written is a bit bland, I would suggest the authors to emphasis more on the outcomes of their study and the knowledge gained. As written, it is hard to grasp what was actually gained from doing this study.
L24 : what is semi ad libitum?
L64: I would change “alter” to “modify” or “affect”. The use of “alter” give it a negative connotation.
L94: there was probably fibre (polysaccharides) that were present due to the cereals and wheat middlings. What do you mean by polysaccharides here?
Table 1: it would be nice to include digestible calcium content of the diets.
L107: would this pelleting temperature inactivate the probiotics?
L123: can you elaborate on how you grouped the piglets? Based on weight but how?
L150: The time is usually reported in military format (ex: 0630 to 1800). Please change throughout the manuscript.
L156: what is the reference for that equation?
L212: how where NSP measured in the samples? I did not see a method listed for that in the 2.3 section.
L215-216: For the equations, use a lower case subscript instead of an underline (ex: Markerratio, Tidiet, Tiexcreta instead of Marker_ratio etc.)
L225-226: Same comment about the equations here
Table 2: What NSP are those? Total, insoluble, soluble? The method needs to be specified.
L370-392: I liked the discussion here. It could be that probiotics allow to uniformize in between diets. For example, in your case, the diet was well digested so probiotics did not do much, but in the case of a challenge or less digestible diet, they could help bring the nutrient utilization to a good level. But this should be tested to be confirmed.
L410: here it would have been nice to have the detail of the NSP analysis such as soluble and insoluble fraction. The soluble fraction is fermented faster. Even though, you found no effect, it could have added to the discussion.
L444-448: I find the conclusion a bit bland. What are the outcomes of the study? What gaps in the literature did you fill, what new knowledge was gained?
Author Response
We kindly thank you for reviewing our manuscript and your fruitful suggestions. We clarify the revisions per comment below.
What the definition of semi ad libitum was, is now clarified by the addition of “(feeding level 3.2x metabolic body weight)” in L21-32. In addition, it was further clarified in equation number 1 (see Line 178).
“Alter” was changed in L74 to “modify”.
Good point about the polysaccharides, we have now removed polysaccharides from the sentence in L110-112 resulting in “The experimental diets did not contain antibiotics, acidifiers or phytase”
The digestible calcium content was provided in Table 1.
We believe that pelleting did not influence the activity of the probiotics as indicated in Supplementary Table S2 illustrating the CFU recovery rate of the mash and pelleted diets.
We have added the following to L135-138: “Piglets were blocked to replicates based on their weaning weight. Variation in weaning weight was 800 g with the lightest piglets weighing 8.10 kg (i.e. replicate 1) and the heaviest piglets weighing 8.90 kg (i.e. replicate 8).”
Changed the time into the military format in L171 and L186
The reference for the semi ad libitum feeding scheme has now been provided in L177
The NSP content was not analysed but calculated with an equation. We have now clarified this with inserting the equation between L237-238
We have adapted all equations so that we use a lower case subscript.
Adapted by using a lower case subscript
The NSP digestibility’s as stated in Table 2 were based on the equation as stated between L237-238 (equation 3).
We would like to thank you about your kind words regarding the discussion. We have added the following to L433-435: “Though more research is necessary to evaluate whether the multi-strain probiotic would be more beneficial in improving nutrient utilization when piglets are reared under more challenging conditions.”
Although we have not analysed the NSP content, we have added the following to L453-457: “The calculated soluble (usually fermentable) carbohydrate content of the experimental di-ets (i.e. hemicellucose and pectins) was 102 g/kg, whereas the insoluble (usually inert) carbohydrate content (i.e. lignin and cellulose) was 63 g/kg. Piglets however only have a limited fermentation capacity and therefore the inclusion of soluble carbohydrates should be limited in weaned piglet diets [12].”.
We have added the following to the conclusion (L494-495): “…fed a European diet. Furthermore, it seems that the multi-strain probiotic could potentially contribute to a reduced N pollution.”
Round 2
Reviewer 2 Report
Comments and Suggestions for Authors
I do not have further comments.
Reviewer 3 Report
Comments and Suggestions for Authors
I read the response to comments, and I think they authors have well answered the concerns and have improved the manuscript. Thank you and good work!